

# Flowing bosonization in the nonperturbative functional renormalization-group approach

**Romain Daviet[1,2] and Nicolas Dupuis[2⋆]**

**1** Institut für Theoretische Physik, Universität zu Köln, D-50937 Cologne, Germany
**2** Sorbonne Université, CNRS, Laboratoire de Physique Théorique de la Matière Condensée, LPTMC, F-75005 Paris, France

⋆ nicolas.dupuis@sorbonne-universite.fr

## Abstract

Bosonization allows one to describe the low-energy physics of one-dimensional quantum fluids within a bosonic effective field theory formulated in terms of two fields: the "density" field $\varphi$ and its conjugate partner, the phase $\vartheta$ of the superfluid order parameter. We discuss the implementation of the nonperturbative functional renormalization group in this formalism, considering a Luttinger liquid in a periodic potential as an example. We show that in order for $\vartheta$ and $\varphi$ to remain conjugate variables at all energy scales, one must dynamically redefine the field $\vartheta$ along the renormalization-group flow. We derive explicit flow equations using a derivative expansion of the scale-dependent effective action to second order and show that they reproduce the flow equations of the sine-Gordon model (obtained by integrating out the field $\vartheta$ from the outset) derived within the same approximation. Only with the scale-dependent (flowing) reparametrization of the phase field $\vartheta$ do we obtain the standard phenomenology of the Luttinger liquid (when the periodic potential is sufficiently weak so as to avoid the Mott-insulating phase) characterized by two low-energy parameters, the velocity of the sound mode and the renormalized Luttinger parameter.

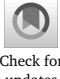

## 1 Introduction

Bosonization is one of the most popular methods to describe one-dimensional quantum fluids [1]. It has recently been used in combination with the nonperturbative functional renormalization (FRG) group to study the Mott-insulating phase induced by a periodic potential (in the framework of the sine-Gordon model) [2,3] and the Bose-glass phase of disordered bosons [4–8]. These studies are based on an action $S[\varphi]$ expressed solely in terms of the "density" field $\varphi$, its conjugate partner, the phase $\vartheta$ of the superfluid order parameter (the field operator $\psi$ for bosons), being integrated out from the outset. In some cases however one would like to keep the field $\vartheta$ in the action in order to study its fluctuations. In other cases, the action is not quadratic in $\vartheta$ and integrating out the latter from the outset in a simple way is not possible.

The basic idea of bosonization is to introduce a low-momentum field $\varphi(x, t)$ such that the long-wavelength part of the density reads $\rho = \rho_0 - \partial_x \varphi/\pi$ where $\rho_0$ is the mean density of particles [9]. The Lagrangian density then reads

$$\mathcal{L}_k = \frac{1}{2\pi K_k} \left[ \frac{1}{v_k}(\partial_t \varphi)^2 - v_k(\partial_x \varphi)^2 \right] + \mathcal{L}_{\text{int},k}. \tag{1}$$

The first two terms on the rhs of (1) can be seen as the leading terms in a derivative expansion and are essentially dictated by symmetries [10]; they yield a mode with linear dispersion $\omega = v_k|q|$. Beside the velocity $v_k$, the Lagrangian density depends on the so-called Luttinger parameter $K_k$ which determines the stiffness $\sim v_k/K_k$ with respect to a local density change. Both $v_k$ and $K_k$ depend on the (coarse-graining) momentum scale $k$ at which the Lagrangian density (1) is defined. The additional contribution $\mathcal{L}_{\text{int},k}$ includes all terms that would be present if $\mathcal{L}_k$ were derived from a microscopic model, e.g. higher-order derivative terms or

various perturbations due to a periodic lattice potential, disorder, etc. When the ground state is a Luttinger liquid, $\mathcal{L}_{\text{int},k}$ is irrelevant (in the RG sense) and the low-energy physics is entirely determined by two parameters, the renormalized velocity $v_R = \lim_{k\to 0} v_k$ and the renormalized Luttinger parameter $K_R = \lim_{k\to 0} K_k$.

In addition to the field $\varphi$, it is possible to consider its conjugate partner[1] $\Pi(x,t) = \partial\mathcal{L}_k/\partial(\partial_t\varphi(x,t))$, which yields the Hamiltonian density $\mathcal{H}_k = \Pi\partial_t\varphi - \mathcal{L}_k$ and the action

$$S_k[\varphi,\vartheta] = \int dt \int dx \left\{ \frac{1}{\pi}\partial_x\vartheta\partial_t\varphi - \frac{v_k}{2\pi}\left[ K_k(\partial_x\vartheta)^2 + \frac{1}{K_k}(\partial_x\varphi)^2 \right] + \mathcal{L}_{\text{int},k} \right\}, \qquad (2)$$

where the field $\vartheta$ is defined by $\Pi = \partial_x\vartheta/\pi$. In the quantum theory, the commutation relation $[\hat{\varphi}(x),\hat{\Pi}(x')] = i\delta(x-x')$ implies $[-\frac{1}{\pi}\partial_x\hat{\varphi}(x),\hat{\vartheta}(x')] = [\hat{\rho}(x),\hat{\vartheta}(x')] = i\delta(x-x')$, which identifies $\vartheta$ as the phase of the field operator (bosons) or the superconducting order parameter (fermions) [1]. This qualitative discussion of bosonization clearly shows that the definition of the field $\vartheta$ is scale dependent; this is a necessary condition to ensure that $\partial_x\varphi$ and $\vartheta$ remain conjugate variables at all scales.[2] This implies that in a RG approach, where the family of actions $S_k[\varphi,\vartheta]$ is obtained from the flow equation $\partial_k S_k[\varphi,\vartheta]$, one has to dynamically redefine the field $\vartheta$ along the flow. The aim of this paper is to show how this can be implemented in the framework of the nonperturbative FRG, considering the case of a Luttinger liquid in a periodic potential as an example. When the field $\vartheta$ is integrated out from the outset, one obtains the sine-Gordon model for which the nonperturbative FRG approach has been shown to be very efficient [2]. In particular the FRG predicts the mass of the lowest excitation (solitons, antisolitons or breathers) with a very good accuracy.

The preservation of the canonical commutation relations between $\partial_x\varphi$ and $\vartheta$ along the RG flow turns out to be crucial for a proper physical description of the system, in particular for the identification of the stiffness of the phase $\vartheta$ as the superfluid density. This differs from the study of a Bose fluid in a periodic potential using the canonically conjugated variables defined by the creation of annihilation boson fields. In that case, the fields $\psi$ and $\psi^*$ defined at the microscopic scale yield a simple identification of the superfluid density in the low-energy limit even though their canonical commutation relations are not preserved along the RG flow (see, e.g., Refs. [11,12] for an FRG study of the Bose-Hubbard model in two and three dimensions).

The outline of the manuscript is as follows. In Sec. 2 we derive the FRG flow equations to second order in a derivative expansion satisfied by the scale-dependent effective action $\Gamma_k[\phi,\theta]$, where $\phi = \langle\varphi\rangle$ and $\theta = \langle\vartheta\rangle$ with $\varphi,\vartheta$ denoting the fields at some initial scale $k_{\text{in}}$. While this approach reproduces the flow equations of the sine-Gordon model and predicts many low-energy properties correctly, it meets with two difficulties: i) When the ground state is a Luttinger liquid, in the limit $k \to 0$ the effective action $\Gamma_k[\phi,\theta]$ is not parametrized by only two parameters, a renormalized velocity $v_R$ and a renormalized Luttinger parameter $K_R$, as expected; ii) the superfluid stiffness does not renormalize despite the presence of a periodic potential. We show that the latter result is a consequence of gauge invariance and therefore independent of the approximation scheme used to solve the FRG flow equations.

In Sec. 3 we show how these issues can be overcome by reparametrizing the field along the RG flow, i.e. by introducing a new field $\bar{\vartheta} \equiv \bar{\vartheta}_k[\varphi,\vartheta]$ defined as a $k$-dependent functional of $\varphi$ and $\vartheta$.[3] This change of variable can be seen as a local frame transformation on

---

[1]This is actually mandatory if one wants to express the field operator $\psi$ in terms of bosonic fields [1].

[2]When $\mathcal{L}_{\text{int},k}$ corresponds to a periodic potential, it is easy to see using perturbation theory that the action $S_{k'}[\varphi,\vartheta]$ (with $k' < k$) contains a term $(\partial_t\varphi)^2$ in addition to $\partial_x\vartheta\partial_t\varphi$. Thus if $\vartheta$ is conjugate to $\varphi$ at scale $k$, this is no longer true at scale $k' < k$.

[3]For previous works using a scale-dependent field reparametrization in the FRG approach, see Refs. [13–22]. The field reparametrization therein is used to eliminate a two-fermion interaction at the expense of an interaction with a collective bosonic field. Although this scale-dependent Hubbard-Stratonovich transformation is referred to

configuration space [25], which can be implemented in two different, but equivalent, ways: as an active frame transformation at the level of the functional integral, or as a passive frame transformation consisting of a mere change of variables $\theta \to \bar{\theta} \equiv \bar{\theta}_k[\phi, \theta]$ in the effective action $\Gamma_k[\phi, \theta]$. We show that to second order of the derivative expansion a linear change of variables is sufficient to ensure that $\bar{\vartheta}$ remains conjugate to $\varphi$ at all scales. In the presence of an external gauge field, in addition to the field reparametrization it is convenient to perform a $k$-dependent gauge transformation $A_\mu \to A'_\mu \equiv A'_{\mu,k}[A_\nu]$ such that $A'_\mu$ enters the effective action in the manifestly gauge invariant form $\partial_\mu \bar{\theta} - A'_\mu$. The effective action $\Gamma_k[\phi, \bar{\theta}]$ exhibits all the physical properties expected when $\bar{\theta}$ is interpreted as the phase of the superfluid order parameter. In particular, in the Luttinger-liquid phase we now find that $\Gamma_{k=0}$ is parametrized only by a renormalized velocity $v_R$ and a renormalized Luttinger parameter $K_R$. Furthermore the superfluid stiffness is reduced by the periodic potential and takes the value $\rho_s = v_R K_R/\pi$ in the infrared limit whereas the compressibility is given by $\kappa = K_R/\pi v_R$.

## 2 FRG and bosonization

We consider a one-dimensional quantum fluid in the presence of a periodic potential. In the bosonization formalism, the low-energy Hamiltonian is given by [1]

$$\hat{H} = \int dx \left\{ \frac{v}{2\pi} \left[ \frac{1}{K} (\partial_x \hat{\varphi}^2) + K (\partial_x \hat{\vartheta})^2 \right] - u \cos(2\sqrt{2}\hat{\varphi}) \right\}, \tag{3}$$

where $\hat{\varphi}$ and $\hat{\vartheta}$ satisfy the commutation relations $[\hat{\vartheta}(x), \partial_y \hat{\varphi}(y)] = i\pi\delta(x - y)$. $\hat{\varphi}$ is related to the density operator *via*

$$\hat{\rho}(x) = \rho_0 - \frac{1}{\pi} \partial_x \hat{\varphi} + 2\rho_2 \cos[2\pi\rho_0 x - 2\hat{\varphi}(x)] + \cdots, \tag{4}$$

where $\rho_0$ is the average density and $\rho_2$ a nonuniversal quantity that depends on microscopic details. The ellipsis in (4) denotes higher-order, subleading, oscillating terms. When $u = 0$ the Hamiltonian (3) describes a Luttinger liquid; $v$ is the the sound-mode velocity and $K$ the Luttinger parameter. The last term in (3) originates from a potential which couples to the density of particles and whose period is commensurate with $1/\rho_0$ [1].[4]

In the functional integral formalism, one obtains the Euclidean (imaginary-time) action

$$S[\varphi, \vartheta] = \int_X \left\{ \frac{v}{2\pi} \left[ \frac{1}{K} (\partial_x \varphi^2) + K (\partial_x \vartheta)^2 \right] - \frac{i}{\pi} \partial_x \varphi \partial_\tau \vartheta - u \cos(2\sqrt{2}\varphi) \right\}, \tag{5}$$

where we use the notation $X = (x, \tau)$ and $\int_X = \int dx \int_0^\beta d\tau$. $\varphi(X)$ and $\vartheta(X)$ are bosonic fields with $\tau \in [0, \beta]$. The model is regularized by a UV cutoff $\Lambda$ acting on both momenta and frequencies. We shall only consider the zero-temperature limit $\beta = 1/T \to \infty$.

### 2.1 Scale-dependent effective action $\Gamma_k[\phi, \theta]$

The strategy of the nonperturbative RG approach is to build a family of models indexed by a momentum scale $k$ such that fluctuations are smoothly taken into account as $k$ is lowered

---

as flowing "bosonization", it has little to do with the flowing bosonization discussed in the present manuscript. Scale-dependent field reparametrization has also been used in Refs. [23, 24] to interpolate between the Cartesian and phase-amplitude representations of the boson field in superfluid systems.

[4]In the case of electrons, Eq. (3) describes only the charge degrees of freedom. Because of spin-charge separation the spin degrees of freedom are insensitive to the periodic potential. The factor $2\sqrt{2}$ in the cosine term would be 2 for bosons; this only modifies the critical value of the Luttinger parameter at the Mott transition ($K_c = 1$ for spin-$\frac{1}{2}$ fermions, 2 for spin-zero bosons).

from a UV scale $k_{\rm in} \gg \Lambda$ down to 0 [26–29]. This is achieved by adding to the action (5) the infrared regulator term

$$\Delta S_k[\varphi, \vartheta] = \frac{1}{2} \sum_Q \big(\varphi(-Q), \vartheta(-Q)\big) R_k(Q) \begin{pmatrix} \varphi(Q) \\ \vartheta(Q) \end{pmatrix}, \tag{6}$$

where $Q = (q, i\omega)$ with $\omega \equiv \omega_n = 2\pi n/\beta$ ($n$ integer) a Matsubara frequency. The cutoff function $R_k(Q)$ is a $2\times 2$ matrix and is chosen so that fluctuation modes satisfying $|q|, |\omega|/v_k \ll k$ are suppressed while those with $|q| \gg k$ or $|\omega|/v_k \gg k$ are left unaffected ($v_k$ is the renormalized velocity of the sound mode); its precise form will be given below.

The partition function

$$\mathcal{Z}_k[J_\varphi, J_\vartheta] = \int \mathcal{D}[\varphi, \vartheta] e^{-S[\varphi, \vartheta] - \Delta S_k[\varphi, \vartheta] + \int_X (J_\varphi \varphi + J_\vartheta \vartheta)} \tag{7}$$

thus becomes $k$ dependent. The expectation values of the fields are given by

$$\phi(X) = \frac{\delta \ln \mathcal{Z}_k[J_\varphi, J_\vartheta]}{\delta J_\varphi(X)} = \langle \varphi(X) \rangle,$$

$$\theta(X) = \frac{\delta \ln \mathcal{Z}_k[J_\varphi, J_\vartheta]}{\delta J_\vartheta(X)} = \langle \vartheta(X) \rangle. \tag{8}$$

The scale-dependent effective action

$$\Gamma_k[\phi, \theta] = -\ln \mathcal{Z}_k[J_\varphi, J_\vartheta] + \int_X (J_\varphi \phi + J_\vartheta \theta) - \Delta S_k[\phi, \theta] \tag{9}$$

is defined as a modified Legendre transform which includes the subtraction of $\Delta S_k[\phi, \theta]$. Assuming that for $k = k_{\rm in}$ the fluctuations are completely frozen by the term $\Delta S_{k_{\rm in}}$ (which is the case when $k_{\rm in}/\Lambda \to \infty$), $\Gamma_{k_{\rm in}}[\phi, \theta] = S[\phi, \theta]$. On the other hand, the effective action of the original model (5) is given by $\Gamma_{k=0}$ provided that $R_{k=0}$ vanishes. The nonperturbative FRG approach aims at determining $\Gamma_{k=0}$ from $\Gamma_{k_{\rm in}}$ using Wetterich's equation [30–32],

$$\partial_t \Gamma_k[\phi, \theta] = \frac{1}{2} \text{Tr} \left\{ \partial_t R_k \big( \Gamma_k^{(2)}[\phi, \theta] + R_k \big)^{-1} \right\}, \tag{10}$$

where $\Gamma_k^{(2)}$ is the second-order functional derivative of $\Gamma_k$ and $t = \ln(k/k_{\rm in})$ a RG "time". The trace in (10) involves a sum over momenta and frequencies.

## 2.2 Derivative expansion and flow equations

In the derivation expansion to second order, the scale-dependent effective action is approximated by

$$\Gamma_k[\phi, \theta] = \int_X \left\{ U_k(\phi) + \frac{1}{2} Z_{1x,k}(\phi)(\partial_x \phi)^2 + \frac{1}{2} Z_{1\tau,k}(\phi)(\partial_\tau \phi)^2 \right.$$
$$\left. + \frac{1}{2} Z_{2,k}(\phi)(\partial_x \theta)^2 - i Z_{3,k}(\phi) \partial_x \phi \, \partial_\tau \theta \right\}, \tag{11}$$

with the initial conditions

$$U_{k_{\rm in}}(\phi) = -u \cos(2\sqrt{2}\phi), \quad Z_{1x,k_{\rm in}}(\phi) = \frac{v}{\pi K}, \quad Z_{1\tau,k_{\rm in}}(\phi) = 0,$$
$$Z_{2,k_{\rm in}}(\phi) = \frac{vK}{\pi}, \quad Z_{3,k_{\rm in}}(\phi) = \frac{1}{\pi}. \tag{12}$$

Note that the terms $\partial_x\phi\partial_x\theta$ et $\partial_\tau\phi\partial_\tau\theta$ are not allowed for symmetry reasons[5] whereas the term $(\partial_\tau\theta)^2$ is forbidden by gauge invariance (Appendix B). The truncation (11) leads to the following two-point vertex for constant, i.e. static and uniform, fields $\phi(X) = \phi$ and $\theta(X) = \theta$,

$$\Gamma_k^{(2)}(Q,\phi) = \begin{pmatrix} Z_{1x,k}(\phi)q^2 + Z_{1\tau,k}(\phi)\omega^2 + U_k''(\phi) & iZ_{3,k}(\phi)q\omega \\ iZ_{3,k}(\phi)q\omega & Z_{2,k}(\phi)q^2 \end{pmatrix}. \tag{13}$$

Its determinant is given by

$$\det\Gamma_k^{(2)}(Q,\phi) = q^2 Z_{1x,k}(\phi)Z_{2,k}(\phi)\left[\frac{\omega^2}{v_k(\phi)^2} + q^2 + \frac{U_k''(\phi)}{Z_{1x,k}(\phi)}\right], \tag{14}$$

with the velocity $v_k(\phi)$ defined by

$$v_k(\phi) = \left(\frac{Z_{1x,k}(\phi)Z_{2,k}(\phi)}{Z_{1\tau,k}(\phi)Z_{2,k}(\phi) + Z_{3,k}(\phi)^2}\right)^{1/2}. \tag{15}$$

The actual velocity is $v_k \equiv v_k(\phi = 0)$, since the minimum of the effective potential corresponds to $\phi = 0$.

We construct the regulator function $R_k(Q)$ by adapting the usual procedure [2, 29] to the two-field formalism used here. We remove $U_k''(\phi)$ from $\Gamma_k^{(2)}(Q,\phi)$, take the average over $\phi$ and multiply the resulting $2 \times 2$ matrix by a function $r$ that freezes the low-energy modes in the functional integral. This gives

$$R_k(Q) = \begin{pmatrix} Z_{1,k}q^2 + \dfrac{Z_{1,k}}{v_k^2}Z_{1\tau,k}\omega^2 & i\dfrac{\sqrt{Z_{1,k}Z_{2,k}}}{v_k}Z_{3,k}q\omega \\ i\dfrac{\sqrt{Z_{1,k}Z_{2,k}}}{v_k}Z_{3,k}q\omega & Z_{2,k}q^2 \end{pmatrix} r\left(\frac{q^2}{k^2} + \frac{\omega^2}{v_k^2 k^2}\right), \tag{16}$$

where

$$Z_{1,k} = \langle Z_{1x,k}(\phi)\rangle_\phi, \qquad Z_{1\tau,k} = \frac{v_k^2}{Z_{1,k}}\langle Z_{1\tau,k}(\phi)\rangle_\phi,$$

$$Z_{2,k} = \langle Z_{2,k}(\phi)\rangle_\phi, \qquad Z_{3,k} = \frac{v_k}{\sqrt{Z_{1,k}Z_{2,k}}}\langle Z_{3,k}(\phi)\rangle_\phi. \tag{17}$$

In practice we take $r(y) = \alpha/(e^y - 1)$. In the case of a precision calculation, e.g. when determining critical exponents at a second-order phase transition [33, 34], $\alpha$ is fixed by using the principle of minimal sensitivity. In the sine-Gordon model, the precise value of $\alpha$ is unimportant [2]. In the present study it is therefore sufficient to take $\alpha$ of order unity. The writing of $R_k(Q)$ in (16) is motivated by the dimensionless variables defined in Appendix A.

Inserting the truncation (11) into Wetterich's equation (10) we obtain coupled flow equations for the functions $U_k(\phi)$, $Z_{1x,k}(\phi)$, $Z_{1\tau,k}(\phi)$, $Z_{2,k}(\phi)$ and $Z_{3,k}(\phi)$ (see Appendix A). We find in particular that $Z_{2,k}(\phi)$ and $Z_{3,k}(\phi)$ do not renormalize, i.e.

$$Z_{2,k}(\phi) = \frac{vK}{\pi}, \qquad Z_{3,k}(\phi) = \frac{1}{\pi}, \tag{18}$$

and

$$v_k(\phi) = v_k = v, \qquad Z_{1\tau,k}(\phi) = \frac{Z_{1x,k}(\phi)}{v^2} - \frac{1}{\pi v K}. \tag{19}$$

In Appendix B we show that Eqs. (18) are a consequence of gauge invariance. The absence of renormalization of the velocity shows that the Lorentz invariance (which is obvious in the sine-Gordon model), i.e. the SO(2) spacetime invariance of the Euclidean action, is preserved by the truncation (11).

---

[5]$\phi$ is odd (even) under parity (time reversal), the reverse being true for $\theta$. Thus the only terms $\partial_\mu\phi\partial_\nu\theta$ being even under parity and time reversal are $\partial_x\phi\partial_\tau\theta$ and $\partial_\tau\phi\partial_x\theta$; these two terms are equivalent when $Z_{3,k}(\phi)$ is independent of $\phi$ (see below, Eq. (18)).

## 2.3 Recovering the flow equations of the sine-Gordon model

The sine-Gordon model is obtained by integrating out the field $\vartheta$ in the action (5). The corresponding scale-dependent effective action reads [2]

$$\Gamma_k^{\text{SG}}[\phi] = \int_X \left\{ U_k(\phi) + \frac{1}{2} Z_k(\phi) \left[ (\partial_x \phi)^2 + \frac{(\partial_\tau \phi)^2}{v^2} \right] \right\}. \tag{20}$$

The physical properties are determined by the effective potential $U_k(\phi)$ and the propagator

$$G_k(Q, \phi) = \frac{1}{Z_k(\phi)(q^2 + \omega^2/v^2) + U_k''(\phi)} \tag{21}$$

in a constant field $\phi$. In Appendix C we show that the functions $U_k(\phi)$ and $Z_k(\phi)$ in the sine-Gordon model are identical to $U_k(\phi)$ and $Z_{1x,k}(\phi)$ appearing in Eq. (11). By solving numerically the flow equations associated with the effective action (11) we thus obtain two phases: a Luttinger-liquid phase where the dimensionless potential $\tilde{U}_k(\phi) = U_k(\phi)/Z_{1,k}k^2$ vanishes in the limit $k \to 0$ and a Mott phase where it flows to a fixed-point potential $\tilde{U}^*(\phi)$ [2].

## 2.4 Physical properties

### 2.4.1 The Luttinger-liquid phase

The effective potential $U_k(\phi)$ is irrelevant in the Luttinger-liquid phase and can therefore be ignored in the low-energy limit. Furthermore, the functions $Z_{1x,k}(\phi) \simeq Z_{1,k}$ and $Z_{1\tau,k}(\phi) \simeq Z_{1,k}Z_{1\tau,k}/v^2$ become $\phi$ independent as can be seen from the numerical solution of the flow equations (Fig. 1). Defining the renormalized Luttinger parameter $K_k$ by the relation

$$Z_{1,k} = \frac{v}{\pi K_k}, \tag{22}$$

the effective action can then be written as

$$\Gamma^{\text{LL}}[\phi, \theta] = \int_X \left\{ \frac{v}{2\pi K_R} (\partial_x \phi)^2 + \frac{Z_{1\tau}}{2\pi v K_R} (\partial_\tau \phi)^2 + \frac{vK}{2\pi} (\partial_x \theta)^2 - \frac{i}{\pi} \partial_x \phi \, \partial_\tau \theta \right\}, \tag{23}$$

in the limit $k \to 0$, where $K_R = K_{k=0}$ and

$$\frac{Z_{1\tau}}{\pi v K_R} \equiv \frac{Z_{1\tau,k=0}}{\pi v K_{k=0}} = \frac{1}{\pi v} \left( \frac{1}{K_R} - \frac{1}{K} \right). \tag{24}$$

The expression (23) does not reproduce the phenomenology of the Luttinger liquid since the latter should be characterized by only two parameters: the velocity $v_R \equiv v$ of the low-energy mode with linear dispersion and a renormalized Luttinger parameter. For $\Gamma^{\text{LL}}$ to describe a Luttinger liquid, we would need $Z_{1\tau}$ to vanish and the coefficient of $(\partial_x \theta)^2$ to depend on $K_R$ instead of $K$. Before introducing the flowing bosonization, which will resolve this issue, we consider the physical properties deduced from the effective action (23). We shall see that the properties that can be obtained from the propagator of the field $\varphi$ (compressibility, density-density response function and conductivity) are correct but the propagator of the field $\vartheta$ and the superfluid stiffness are not (insofar as they do not agree with the expected results in a Luttinger liquid).

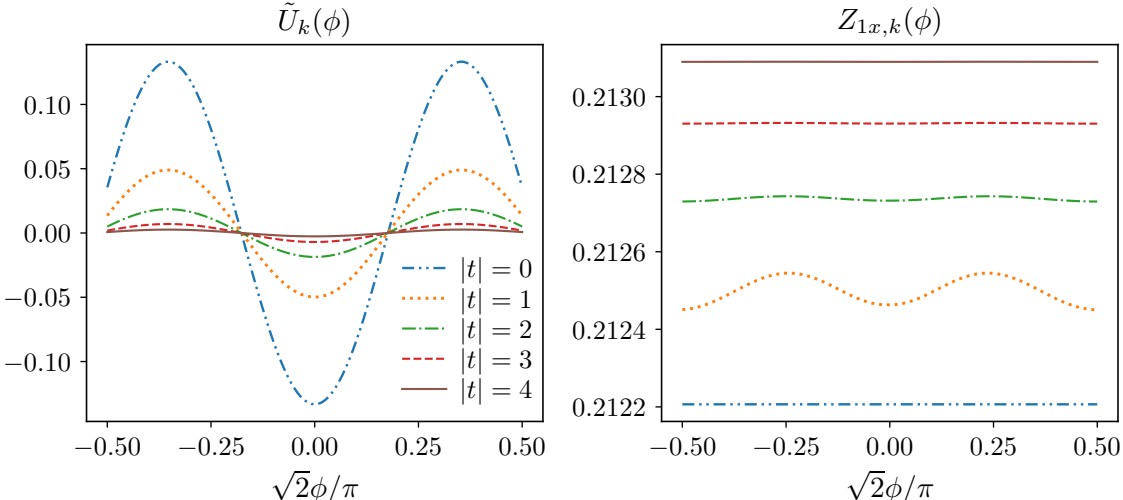

Figure 1: $\tilde{U}_k(\phi) = U_k(\phi)/Z_{1,k}k^2$ and $Z_{1x,k}(\phi)$ in the Luttinger-liquid phase for various values of $k \leq \Lambda$. $Z_{1\tau,k}(\phi)$ is related to $Z_{1x,k}(\phi)$ by Eq. (19). In Figs. 1 and 2, $t = \ln(k/\Lambda)$ denotes the (negative) RG time. ($\Lambda = 1$, $u/\Lambda^2 = 0.01$ and $K = 1.5$.)

**Density-density correlation function.** By inverting the two-point vertex (13) in the Luttinger-liquid phase, we obtain the propagator of the field $\varphi$,

$$G_{\varphi\varphi}(Q) = \langle \varphi(Q)\varphi(-Q) \rangle = \frac{\pi v K_R}{\omega^2 + v^2 q^2}, \tag{25}$$

which is the standard expression in a Luttinger liquid with parameters $v$ and $K_R$. Using $\rho = \rho_0 - \partial_x \varphi/\pi$ in the long-wavelength limit, we deduce the density-density correlation function

$$\chi_{\rho\rho}(Q) = \frac{q^2}{\pi^2} G_{\varphi\varphi}(Q) = \frac{v K_R}{\pi} \frac{q^2}{\omega^2 + v^2 q^2} \qquad (|q| \ll 1/\rho_0) \tag{26}$$

and the compressibility

$$\kappa = \lim_{q \to 0} \chi_{\rho\rho}(q, i\omega = 0) = \frac{K_R}{\pi v}. \tag{27}$$

The conductivity can be obtained from its relation to the density-density correlation function (which follows from gauge invariance) [35]

$$\sigma(\omega) = \lim_{q \to 0} \frac{-i\omega}{q^2} \chi_{\rho\rho}(q, \omega + i0^+) = \frac{v K_R}{\pi} \frac{i}{\omega + i0^+}, \tag{28}$$

which leads to a Drude weight (defined as the weight of the Dirac peak $\delta(\omega)$ in $\sigma(\omega)$) $D = v K_R$. Equations (26-28) reproduce the known results in a Luttinger liquid [1].

**Current-current correlation function.** The conductivity can also be obtained from

$$\sigma(\omega) = -\frac{i}{\omega + i0^+} K_{xx}(0, \omega + i0^+), \tag{29}$$

where $K_{xx}$ is the response to an external vector potential $A_x$,

$$K_{xx}(Q) = \langle j_x(Q) j_x(-Q) \rangle - \frac{v K}{\pi} = \left(\frac{v K}{\pi}\right)^2 q^2 G_{\vartheta\vartheta}(Q) - \frac{v K}{\pi}. \tag{30}$$

The last term in this expression corresponds to the diamagnetic contribution while $j_x = (vK/\pi)\partial_x\vartheta$ is the paramagnetic part of the current. Using (23) we obtain

$$G_{\vartheta\vartheta}(Q) = \frac{\pi}{vK}\frac{v^2q^2 + (1-K_R/K)\omega^2}{q^2(\omega^2 + v^2q^2)}, \tag{31}$$

and

$$K_{xx}(Q) = -\frac{vK_R}{\pi}\frac{\omega^2}{\omega^2 + v^2q^2}. \tag{32}$$

Although the propagator $G_{\vartheta\vartheta}$ takes an unusual expression, Eq. (32) is the usual result for a Luttinger liquid. Equation (29) agrees with (28).

**Superfluid stiffness.** The superfluid stiffness $\rho_s$ is defined by

$$G_{\vartheta\vartheta}(q, i\omega = 0) = \frac{1}{\rho_s q^2} \qquad (q \to 0) \tag{33}$$

or, equivalently, by the coefficient of the term $(\partial_x\theta)^2$ in the effective action. We thus find that $\rho_s = vK/\pi$ is not renormalized, an unexpected feature in the presence of a periodic potential, which is related to the unusual expression of the propagator (31) and is a consequence of gauge invariance (Appendix B).

### 2.4.2 The Mott-insulating phase

Since the flow equations reproduce those of the sine-Gordon model, all physical quantities that can be deduced from the effective potential $U_k(\phi)$ or the propagator $G_{\varphi\varphi,k}(Q,\phi)$ are identical to those in the sine-Gordon model. This means in particular that the mass of the solitons and antisolitons, as well as the mass of the lowest soliton-antisoliton bound state (breather), are obtained with a very good accuracy [2].

In the Mott-insulating phase the propagator $G_{\varphi\varphi,k}$ evaluated at vanishing field $\phi = 0$ (which corresponds to the minimum of the effective potential and therefore to the physical state) is given by

$$G_{\varphi\varphi,k}(Q) = \frac{1}{Z_{1x,k}(0)[q^2 + (\omega^2 + m_k^2)/v^2]}, \tag{34}$$

where

$$m_k = v\sqrt{\frac{U_k''(0)}{Z_{1x,k}(0)}} \tag{35}$$

is the mass of the lowest excitation in the topological sector $Q = 0$: a pair soliton-antisoliton for $1/2 \leq K \leq 1$ ($m_k \to m_{\text{sol}} + m_{\text{antisol}} = 2m_{\text{sol}}$ for $k \to 0$) or a soliton-antisoliton bound state for $K \leq 1/2$ ($m_k \to m_{\text{breather}}$) [2]. Because of the nonzero mass $m = \lim_{k\to 0} m_k$, the compressibility vanishes and the optical conductivity is gapped, as expected for a Mott insulator.

The propagator of the field $\vartheta$ reads

$$G_{\vartheta\vartheta,k}(Q) = \frac{\pi}{vK}\frac{v^2q^2 + m_k^2 + (1-K_k/K)\omega^2}{q^2(\omega^2 + v^2q^2 + m_k^2)}, \tag{36}$$

where the renormalized Luttinger parameter $K_k$ is now defined from the vanishing field configuration (which corresponds to the physical state),

$$Z_{1x,k}(0) = \frac{v}{\pi K_k}. \tag{37}$$

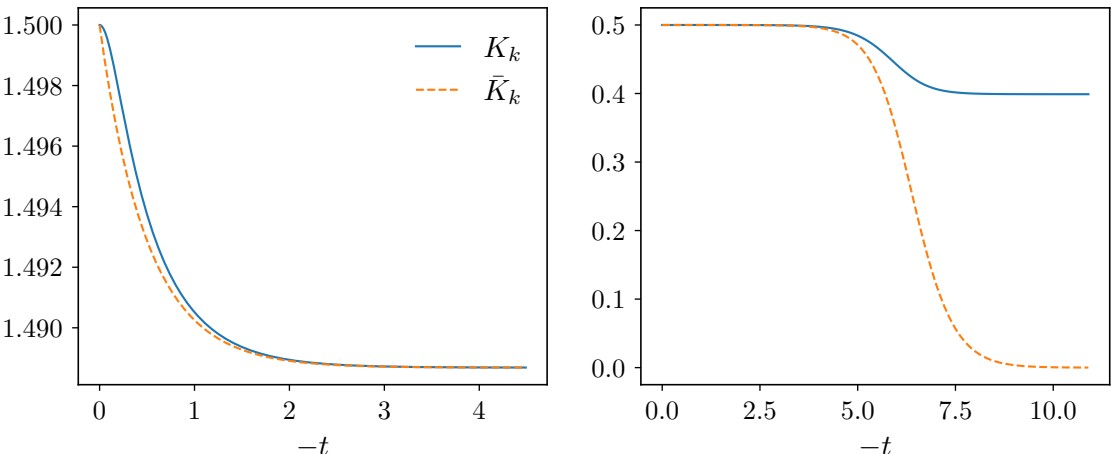

Figure 2: $K_k = \nu/\pi Z_{1x,k}(0)$ and $\bar{K}_k = \nu/\pi Z_{1,k}$ vs $k$ in the Luttinger-liquid phase with $\Lambda = 1$, $u/\Lambda^2 = 0.01$ and $K = 1.5$ (left) and the Mott-insulating phase with $\Lambda = 1$, $u/\Lambda^2 = 0.001$ and $K = 0.5$ (right).

Note that this definition differs from the one used in Ref. [2] where the Luttinger parameter, which we denote here by $\bar{K}_k$, was deduced from the field average of $Z_{1x,k}(\phi)$, i.e. $Z_{1,k} = \nu/\pi \bar{K}_k$. $K_k$ and $\bar{K}_k$ coincide in the Luttinger-liquid phase when $k \to 0$ but differ in the Mott-insulating phase: Whereas $\bar{K}_k$ vanishes, $K_k$ remains finite (Fig. 2). Again we see that the superfluid stiffness does not renormalize, in agreement with the conclusion of Appendix B, since Eqs. (33) and (36) yield $\rho_s = \nu K/\pi$.

## 3 Flowing bosonization

As argued in the introduction, a proper definition of the phase field $\vartheta$ (and therefore $\theta$) should be scale dependent. The difficulties encountered in the FRG approach described in Sec. 2, in particular the absence of renormalization of the superfluid stiffness, are therefore not surprising. In this section, we show how the FRG approach can be implemented with a scale-dependent field $\bar{\vartheta} \equiv \bar{\vartheta}_k[\varphi, \vartheta]$, defined as a $k$-dependent functional of $\varphi$ and $\vartheta$, such that $\varphi$ and $\bar{\vartheta}$ remain conjugate variables at all scales. This can be most simply achieved by considering the $k$-dependent change of variable $\theta \to \bar{\theta} \equiv \bar{\theta}_k[\phi, \theta]$ and expressing the effective action $\Gamma_k[\phi, \theta_k[\phi, \bar{\theta}]]$ in terms of the new variables. Here we assume the map $\bar{\theta}_k[\phi, \theta]$ to be invertible so that the inverse map $\theta_k[\phi, \bar{\theta}]$ exists. In the language of Ref. [25], such a change of variable can be seen as a passive frame transformation. Alternatively, one can consider an active frame transformation, where the change of variable $\vartheta \to \bar{\vartheta} \equiv \bar{\vartheta}_k[\varphi, \vartheta]$ is performed at the level of the functional integral, and compute the effective action $\bar{\Gamma}_k[\phi, \bar{\theta}]$, where $\bar{\theta} = \langle \bar{\vartheta} \rangle$, from its flow equation. The passive and active points of view lead to the same effective action, $\Gamma_k[\phi, \theta_k[\phi, \bar{\theta}]] = \bar{\Gamma}[\phi, \bar{\theta}]$, a consequence of the linear nature of the transformation between the fields used here [25].

### 3.1 Passive frame transformation

#### 3.1.1 Effective action $\Gamma_k[\phi, \theta_k[\phi, \bar{\theta}]]$

The phase field $\bar{\theta}$ is defined by

$$\theta(Q) = \alpha_k(Q)\phi(Q) + \beta_k(Q)\bar{\theta}(Q), \tag{38}$$

where we assume $\alpha_k(-Q) = \alpha_k(Q)$ and $\beta_k(Q) = \beta_k(-Q)$. The effective action becomes

$$\Gamma_k[\phi, \theta_k[\phi, \bar{\theta}]] = \int_X \left\{ U_k(\phi) + \frac{1}{2} Z_{1x,k}(\phi)(\partial_x \phi)^2 + \frac{1}{2} Z_{1\tau,k}(\phi)(\partial_\tau \phi)^2 \right\}$$
$$+ \sum_Q \left\{ \left[ \frac{vK}{2\pi} q^2 \alpha_k(Q)^2 + \frac{i}{\pi} q\omega \alpha_k(Q) \right] \phi(-Q)\phi(Q) \right.$$
$$\left. + \left[ \frac{vK}{\pi} q^2 \alpha_k(Q) + \frac{i}{\pi} q\omega \right] \beta_k(Q)\phi(-Q)\bar{\theta}(Q) + \frac{vK}{2\pi} q^2 \beta_k(Q)^2 \bar{\theta}(-Q)\bar{\theta}(Q) \right\}. \quad (39)$$

The coefficients $\alpha_k(Q)$ and $\beta_k(Q)$ are determined by requiring that the coupling between $\phi$ and $\bar{\theta}$ be of the form $-\frac{i}{\pi}\partial_x\phi\partial_\tau\bar{\theta}$ as well as the absence of term $(\partial_\tau\phi)^2$; these conditions indeed ensure that the two fields are conjugate variables. The latter condition cannot be satisfied for all values of $\phi$ and we shall therefore impose it only for $\phi = 0$. In the Mott phase, $\phi = 0$ corresponds to the physical state whereas in the Luttinger-liquid phase the vanishing of the term $(\partial_\tau\phi)^2$ will be satisfied for all fields since $Z_{1\tau,k}(\phi)$ becomes $\phi$ independent when $k \to 0$. We thus obtain the equations

$$\frac{1}{2} Z_{1\tau,k}(0)\omega^2 + \frac{vK}{2\pi} q^2 \alpha_k(Q)^2 + \frac{i}{\pi} q\omega \alpha_k(Q) = 0,$$
$$\left[ \frac{vK}{\pi} q^2 \alpha_k(Q) + \frac{i}{\pi} q\omega \right] \beta_k(Q) = \frac{i}{\pi} q\omega. \quad (40)$$

Demanding that $\alpha_k(Q) = 0$ when $K_k = K$, we find

$$\alpha_k(Q) = -\frac{i}{K} \frac{\omega}{vq} \left( 1 - \sqrt{\frac{K}{K_k}} \right), \qquad \beta_k(Q) = \sqrt{\frac{K_k}{K}}, \quad (41)$$

where the scale-dependent Luttinger parameter $K_k$ is defined by (37). The change of variables (38) can be rewritten in the insightful form

$$\frac{vK}{\pi} \partial_x \theta = \left( 1 - \sqrt{\frac{K}{K_k}} \right) \frac{i}{\pi} \partial_\tau \phi + \sqrt{\frac{K}{K_k}} \frac{vK_k}{\pi} \partial_x \bar{\theta}. \quad (42)$$

This amounts to rewriting the expectation value of the current $\langle j_\vartheta \rangle = (vK/\pi)\partial_x\theta$ as a weighted sum of $\langle j_\varphi \rangle = (i/\pi)\partial_\tau\phi$ and $\langle j_{\bar{\vartheta}} \rangle = (vK_k/\pi)\partial_x\bar{\theta}$. $j_\varphi = (i/\pi)\partial_\tau\varphi$ is the expression of the current in the sine-Gordon model[6] whereas $j_{\bar{\vartheta}}$ is the standard expression of the current associated with the phase field $\bar{\theta}$ and the stiffness $vK_k/\pi$. Equation (39) confirms that the stiffness associated with $\bar{\theta}$, defined as the coefficient of $(\partial_x\bar{\theta})^2$, i.e.

$$\frac{vK}{\pi} \beta_k(Q)^2 = \frac{vK_k}{\pi}, \quad (43)$$

depends on the renormalized Luttinger parameter $K_k$.

We thus obtain the following expression of the effective action,

$$\Gamma_k[\phi, \theta_k[\phi, \bar{\theta}]] = \int_X \left\{ U_k(\phi) + \frac{1}{2} Z_{1x,k}(\phi)(\partial_x \phi)^2 + \frac{1}{2} [Z_{1\tau,k}(\phi) - Z_{1\tau,k}(0)](\partial_\tau \phi)^2 \right.$$
$$\left. - \frac{i}{\pi} \partial_x \phi \partial_\tau \bar{\theta} + \frac{vK_k}{2\pi} (\partial_x \bar{\theta})^2 \right\}. \quad (44)$$

---

[6]The expression of the current $j_\varphi$ follows from the continuity equation $\partial_t\rho + \partial_x j_\varphi = 0$ with $\rho = \rho_0 - \frac{1}{\pi}\partial_x\varphi$ and $t = -i\tau$.

In the Luttinger-liquid phase, where $Z_{1x,k}(\phi)$ and $Z_{1\tau,k}(\phi)$ becomes $\phi$ independent in the limit $k \to 0$ and $U_k(\phi)$ is irrelevant, one has

$$\Gamma_{k=0}^{\text{LL}}[\phi, \theta_{k=0}[\phi, \bar{\theta}]] = \int_X \left\{ \frac{v}{2\pi} \left[ \frac{1}{K_R}(\partial_x \phi)^2 + K_R(\partial_x \bar{\theta})^2 \right] - \frac{i}{\pi} \partial_x \phi \partial_\tau \bar{\theta} \right\}. \tag{45}$$

This is the usual effective action of a Luttinger liquid characterized by the velocity $v$ of its low-energy mode and the Luttinger parameter $K_R = K_{k=0}$. From (45) we recover the expression of the density-density response function (26). We can compute the response function $K_{xx}(Q)$ using

$$j_\vartheta = \left( 1 - \sqrt{\frac{K}{K_k}} \right) \frac{i}{\pi} \partial_\tau \varphi + \sqrt{\frac{K}{K_k}} \frac{v K_k}{\pi} \partial_x \bar{\vartheta}, \tag{46}$$

which is the analog of (42) but for the fields $\vartheta$, $\varphi$ and $\bar{\vartheta}$. This gives

$$K_{xx}(Q) = -\left( 1 - \sqrt{\frac{K}{K_R}} \right)^2 \frac{\omega^2}{\pi^2} G_{\varphi\varphi}(Q) - \frac{2i}{\pi^2} v K_R \sqrt{\frac{K}{K_R}} \left( 1 - \sqrt{\frac{K}{K_R}} \right) \omega q G_{\varphi\vartheta}(Q)$$
$$+ \left( \frac{v K_R}{\pi} \right)^2 \frac{K}{K_R} q^2 G_{\vartheta\vartheta}(Q) - \frac{v K}{\pi} \tag{47}$$

and, using the expression of the propagators deduced from (45), we reproduce (32). Although Eq. (47) is correct, it takes a somewhat unsatisfying form since it involves both the bare stiffness $K$ and the renormalized one $K_R$. We shall see in the next section how we can express the electromagnetic response function in a more natural way without any reference to the bare stiffness.

In the Mott-insulating phase the term $(\partial_\tau \phi)^2$ does not vanish for all values of the field but the two-point vertex, defined as the matrix of functional derivatives with respect to $\phi$ and $\bar{\theta}$, takes the form

$$\Gamma_k^{(2)}(Q) = \begin{pmatrix} \dfrac{1}{\pi v K_k}(v^2 q^2 + m_k^2) & \dfrac{i}{\pi} q \omega \\ \dfrac{i}{\pi} q \omega & \dfrac{v K_k}{\pi} q^2 \end{pmatrix} \tag{48}$$

in the physical state ($\phi = 0$). The only frequency-dependent term is the coupling between $\phi$ and $\bar{\theta}$ as expected for two conjugate variables. The propagator of the field $\varphi$ in the physical state is still given by (34) and yields a vanishing compressibility and a gapped optical conductivity. The propagator of $\bar{\vartheta}$ differs from (36) and reads

$$G_{\bar{\vartheta}\bar{\vartheta},k}(Q) = \frac{\pi}{v K_k} \frac{v^2 q^2 + m_k^2}{q^2(\omega^2 + v^2 q^2 + m_k^2)}, \tag{49}$$

which gives the superfluid stiffness $\rho_{s,k} = v K_k / \pi$ using the definition (33) but with the propagator $G_{\bar{\vartheta}\bar{\vartheta},k}$. Since $K_k$ decreases with $k$ but does not vanish in the limit $k \to 0$ (Fig. 2), we find that the stiffness remains finite in disagreement with the expected result for a Mott insulator. A possible explanation comes from the convergence of $\eta_{1,k} = -\partial_t \ln Z_{1,k}$ towards 2 which makes the regulator of order $Z_{1,k} k^2 \sim k^{2-\eta_k}$ for $|q|, |\omega|$ of order $k$. Thus the convergence of $\eta_{1,k}$ towards 2 must be extremely slow,[7] which is not realized in practice, for the regulator function $R_k$ to vanish in the infrared [3]. While this issue is irrelevant for most physical quantities, which rapidly converge when $k$ becomes smaller than the mass scale $m_k/v$, the non-vanishing of $R_k$ may artificially stop the flow of $K_k$ thus preventing the superfluid stiffness to vanish when $k \to 0$.

---

[7]Note that $\lim_{k \to 0} \eta_{1,k} = 2$ implies that $\bar{K}_k$ vanishes as $k^2$.

Beyond the second order of the derivative expansion one expects additional terms in the effective action, such as $(\partial_\tau \phi)^4$, incompatible with $\partial_x \varphi$ and $\vartheta$ being conjugate fields. The linear change of variable (38) will not allow us to cancel all these terms while keeping the coefficient of $\partial_x \phi \partial_\tau \bar{\theta}$ equal to $-i/\pi$. Whether this could be achieved with a nonlinear change of variables is an open issue.

### 3.1.2 Coupling to an external gauge field

In the presence of an external gauge field $A_\mu$ ($\mu = 0, x$), gauge invariance implies that the effective action $\Gamma_k[\phi, \theta, A]$ is simply deduced from $\Gamma_k[\phi, \theta]$ by replacing $\partial_\mu \theta$ by the covariant derivative $\partial_\mu \theta - A_\mu$. The change of variables (38) will not preserve this simple structure. It is however possible to perform a scale-dependent gauge transformation $A_\mu \to A'_\mu$ so that the vector potential $A'_\mu$ enters the effective action $\Gamma_k[\phi, \theta_k[\phi, \bar{\theta}], A'_\mu]$ in the covariant expression $\partial_\mu \bar{\theta} - A'_\mu$.

The effective action $\Gamma_k[\phi, \theta, A]$ can be written as

$$\Gamma_k[\phi, \theta, A] = \Gamma_k[\phi, \theta] + \int_X \left( \frac{vK}{2\pi} A_x^2 - \frac{vK}{\pi} A_x \partial_x \theta + \frac{i}{\pi} A_0 \partial_x \phi \right). \tag{50}$$

Performing the passive frame transformation (38), we obtain

$$\Gamma_k[\phi, \theta_k[\phi, \bar{\theta}], A] = \Gamma_k[\phi, \theta_k[\phi, \bar{\theta}]] + \int_X \left\{ \frac{vK}{2\pi} A_x^2 - A_x \frac{vK}{\pi} \partial_x \theta + \frac{i}{\pi} A_0 \partial_x \phi \right\}, \tag{51}$$

where the current $(vK/\pi)\partial_x \theta$ can be expressed in terms of $\phi$ and $\bar{\theta}$ using (42). We now consider the gauge transformation

$$A'_\mu = A_\mu + \partial_\mu \xi \quad \text{with} \quad \partial_x \xi = \left( \sqrt{\frac{K}{K_k}} - 1 \right) A_x, \tag{52}$$

which leaves the partition function and therefore the effective action $\Gamma_k[\phi, \theta, A] = \Gamma_k[\phi, \theta', A']$ unchanged.[8] Equation (52) implies

$$A_x(Q) = \sqrt{\frac{K_k}{K}} A'_x(Q),$$

$$A_0(Q) = A'_0(Q) + \frac{\omega}{q} \left( 1 - \sqrt{\frac{K_k}{K}} \right) A'_x(Q), \tag{53}$$

and therefore

$$\begin{aligned} \Gamma_k[\phi, \theta_k[\phi, \bar{\theta}], A'_\mu] &= \Gamma_k[\phi, \theta_k[\phi, \bar{\theta}]] + \int_X \left\{ \frac{vK_k}{2\pi} A'^2_x - A'_x \frac{vK_k}{\pi} \partial_x \bar{\theta} + \frac{i}{\pi} A'_0 \partial_x \phi \right\} \\ &= \int_X \left\{ U_k(\phi) + \frac{1}{2} Z_{1x,k}(\phi)(\partial_x \phi)^2 + \frac{1}{2}[Z_{1\tau,k}(\phi) - Z_{1\tau,k}(0)](\partial_\tau \phi)^2 \right. \\ &\quad \left. - \frac{i}{\pi} \partial_x \phi (\partial_\tau \bar{\theta} - A'_0) + \frac{vK_k}{2\pi} (\partial_x \bar{\theta} - A'_x)^2 \right\}. \end{aligned} \tag{54}$$

---

[8] The invariance of the partition function in the transformation (52) also requires the change of variables $\vartheta' = \vartheta + \xi$. The effective action $\Gamma_k[\phi, \theta', A'] = \Gamma_k[\phi, \theta, A]$ becomes a functional of $\theta' = \theta + \xi$. We simply denote $\theta'$ by $\theta$ in the following.

In the gauge $A'_\mu$, the expectation values of the current densities take the usual form,

$$
\begin{aligned}
\langle j_0(X)\rangle &= -\frac{\delta\Gamma_k[\phi,\theta_k[\phi,\bar\theta],A'_\mu]}{\delta A'_0(X)} = -\frac{i}{\pi}\partial_x\phi \equiv i[\langle\rho(X)\rangle - \rho_0],\\
\langle J_x(X)\rangle &= -\frac{\delta\Gamma_k[\phi,\theta_k[\phi,\bar\theta],A'_\mu]}{\delta A'_x(X)} = \frac{\nu K_k}{\pi}\partial_x\bar\theta(X) - \frac{\nu K_k}{\pi} \equiv \langle j_x(X)\rangle - \frac{\nu K_k}{\pi},
\end{aligned}
\tag{55}
$$

where $\rho(X) = \rho_0 - \partial_x\varphi/\pi$ denotes here the long-wavelength part of the density and $j_x = (\nu K_k/\pi)\partial_x\bar\theta$ the paramagnetic part of the current. The diamagnetic contribution to $\langle J_x\rangle$ depends on the renormalized Luttinger parameter $K_k$.

## 3.2 Active frame transformation

In the active frame transformation one considers the new field $\bar\vartheta \equiv \bar\vartheta_k[\varphi,\vartheta]$ defined by

$$
\vartheta(Q) = \alpha_k(Q)\varphi(Q) + \beta_k(Q)\bar\vartheta(Q)
\tag{56}
$$

and the partition function

$$
\mathcal{Z}_k[J_\varphi, J_{\bar\vartheta}] = \int \mathcal{D}[\varphi,\vartheta]\exp\left\{-S[\varphi,\vartheta] - \Delta S_k[\varphi,\vartheta] + \int_X (J_\varphi\varphi + J_{\bar\vartheta}\bar\vartheta)\right\}.
\tag{57}
$$

For a linear change of variables, the active frame transformation (56) is the counterpart of the passive transformation (38) and the coefficients $\alpha_k(Q)$ and $\beta_k(Q)$ are therefore given by (41) [25]. The external source $J_{\bar\vartheta}$ couples to the new field $\bar\vartheta$ so that $\ln\mathcal{Z}_k[J_\varphi, J_{\bar\vartheta}]$ is the generating functional of the connected correlation functions of the fields $\varphi$ and $\bar\vartheta$. Expressing $\Delta S_k[\varphi,\vartheta] = \Delta\bar S_k[\varphi,\bar\vartheta]$ in terms of the new variables, we obtain

$$
\Delta\bar S_k[\varphi,\bar\vartheta] = \frac{1}{2}\sum_Q (\varphi(-Q),\bar\vartheta(-Q))\bar R_k(Q)\begin{pmatrix}\varphi(Q)\\\bar\vartheta(Q)\end{pmatrix},
\tag{58}
$$

where

$$
\bar R_k(Q) = \begin{pmatrix} Z_{1,k}q^2 + \left(\dfrac{1}{\bar K_k} - \dfrac{1}{K_k}\right)\dfrac{\omega^2}{\pi\nu} & \dfrac{i}{\pi}q\omega \\[2ex] \dfrac{i}{\pi}q\omega & \dfrac{\nu K_k}{\pi}q^2 \end{pmatrix} r\left(\frac{q^2}{k^2} + \frac{\omega^2}{\nu^2 k^2}\right)
\tag{59}
$$

and $\bar K_k$ is defined in Sec. 2.4.2. The cutoff function $\bar R_k$ can also be deduced from the effective action $\Gamma_k[\phi,\theta_k[\phi,\bar\theta]]$ obtained in Sec. 3.1 in the same way as $R_k$ was deduced from $\Gamma_k[\phi,\theta]$. Anticipating that $\Gamma_k[\phi,\theta_k[\phi,\bar\theta]]$ is identical to the effective action $\bar\Gamma_k[\phi,\bar\theta]$ defined as the Legendre transform of $\ln\mathcal{Z}_k[J_\varphi, J_{\bar\vartheta}]$, we conclude that $\Delta\bar S_k[\varphi,\bar\vartheta]$ is the natural regulator for the fields $\varphi$ and $\bar\vartheta$.

### 3.2.1 Effective action $\bar\Gamma_k[\phi,\bar\theta]$

The scale-dependent effective action is defined by

$$
\bar\Gamma_k[\phi,\bar\theta] = -\ln\mathcal{Z}_k[J_\varphi, J_{\bar\vartheta}] + \int_X (J_\varphi\phi + J_{\bar\vartheta}\bar\theta) - \Delta\bar S_k[\phi,\bar\theta],
\tag{60}
$$

where

$$
\begin{aligned}
\phi(X) &= \frac{\delta\ln\mathcal{Z}_k[J_\varphi, J_{\bar\vartheta}]}{\delta J_\varphi(X)} = \langle\varphi(X)\rangle,\\
\bar\theta(X) &= \frac{\delta\ln\mathcal{Z}_k[J_\varphi, J_{\bar\vartheta}]}{\delta J_{\bar\vartheta}(X)} = \langle\bar\vartheta(X)\rangle,
\end{aligned}
\tag{61}
$$

and satisfies the flow equation (see Eq. (94) in Appendix D)

$$\partial_k \bar{\Gamma}_k[\phi, \bar{\theta}] = \frac{1}{2} \text{Tr} \Big[ \partial_k \bar{R}_k \big( \bar{\Gamma}_k^{(2)}[\phi, \bar{\theta}] + \bar{R}_k \big)^{-1} \Big] - \int_X \frac{\delta \bar{\Gamma}_k[\phi, \bar{\theta}]}{\delta \bar{\theta}(X)} \langle \partial_k \bar{\vartheta}_k(X) \rangle$$

$$+ \int_{X,Y} [\bar{R}_{\bar{\vartheta}\varphi,k}(X,Y) \langle \partial_k \bar{\vartheta}_k(X) \varphi(Y) \rangle_c + \bar{R}_{\bar{\vartheta}\bar{\vartheta},k}(X,Y) \langle \partial_k \bar{\vartheta}_k(X) \bar{\vartheta}_k(Y) \rangle_c], \quad (62)$$

where $\bar{\vartheta}_k \equiv \bar{\vartheta}_k[\varphi, \vartheta]$ and $\langle \cdots \rangle_c$ denotes a connected correlation function. This equation is a particular case, corresponding to a linear reparametrization of the fields, of the general equation derived in Refs. [19, 21, 25, 36].

### 3.2.2 Equivalence between the active and passive points of view

The effective actions $\Gamma_k[\phi, \theta_k[\phi, \bar{\theta}]]$ and $\bar{\Gamma}_k[\phi, \bar{\theta}]$ are obviously equal for $k = k_{\text{in}}$. In Appendix E we show that they satisfy the same flow equation,

$$\partial_k \bar{\Gamma}_k[\phi, \bar{\theta}] \Big|_{\phi, \bar{\theta}} = \partial_k \Gamma_k[\phi, \theta_k[\phi, \bar{\theta}]] \Big|_{\phi, \bar{\theta}}, \quad (63)$$

so that they are equal for all values of $k$ and given by (44). This equivalence is a consequence of the change of variables $(\varphi, \vartheta) \rightarrow (\varphi, \bar{\vartheta})$ being linear [25].

## 4 Conclusion

The standard FRG study of a Bose fluid in a periodic potential is based on the effective action $\Gamma_k[\psi^*, \psi]$ expressed as a functional of two conjugate variables: the expectation values $\psi$ and $\psi^*$ of the boson field and its conjugate partner [11, 12]. Yet in that case it is not necessary to dynamically redefine the fields along the flow. The superfluid density $\rho_s$, obtained from the coefficient of $|\nabla\psi|^2$ in the effective action, is reduced by quantum fluctuations and is related to the Drude weight by $\rho_s = D/\pi$ in the superfluid phase [37].[9] The dynamical term $\psi^*\partial_\tau\psi$, which is due to $\psi$ and $\psi^*$ being conjugate fields at the microscopic scale, is renormalized and even vanishes in the limit $k \rightarrow 0$ in the superfluid phase whereas a second-order time-derivative term $|\partial_\tau\psi|^2$ is generated [38–41]. On the contrary, in the bosonization framework, to obtain a meaningful description of the superfluid properties it is necessary to redefine the phase field $\vartheta$ so that $\varphi$ and $\vartheta$ remain manifestly conjugate variables. There is no difficulty to implement the field reparametrization in the derivative expansion to second order, a mere linear change of variable being sufficient. The flow equations both reproduce those of the sine-Gordon model and yield a low-energy description of the Luttinger-liquid phase in terms of two parameters, the renormalized velocity $v_R$ of the sound mode and the renormalized Luttinger parameter $K_R$.

A proper treatment of the phase field $\vartheta$ is an important step in the FRG analysis of low-dimensional quantum fluids in the framework of bosonization. For instance, this will allow a more accurate study of the Bose-glass phase of a one-dimensional disordered Bose fluid. The previous works using bosonization and FRG are based on an effective model obtained by integrating out the field $\vartheta$ from the outset [4, 5]. This is sufficient to determine the properties related to the density field and its fluctuations but provides us with little information on the superfluid properties and the correlation function of the phase field $\vartheta$. The work reported in this manuscript also opens up the possibility to study strongly anisotropic two- or three-dimensional systems, consisting of weakly coupled one-dimensional chains. In these systems

---

[9]The relativistic O(2) model studied in Ref. [37] also describes superfluids because of the emergent Lorentz invariance at low energies in these systems [38–40].

the interchain kinetic coupling $\psi_n^* \psi_m \sim e^{-i\vartheta_n + i\vartheta_m}$ depends nontrivially on $\vartheta$ and it is not possible to integrate out this field from the outset. An RG approach must therefore necessarily consider the fields $\varphi$ and $\vartheta$ on equal footing.

## Acknowledgements

We would like to thank Riccardo Ben Alí Zinati, Kevin Falls, Adam Rançon, Patrick Azaria, Philippe Lecheminant and Jan Pawlowski for useful discussions.

## A  Flow equations

### A.1  Dimensionless variables

The flow equations are solved by introducing the dimensionless variables $\tilde{q} = q/k$, $\tilde{\omega} = \omega/v_k k$, where $v_k = v_k(\phi = 0)$ [Eq. (15)], and the dimensionless functions

$$
\tilde{U}_k(\phi) = \frac{U_k(\phi)}{Z_{1,k} k^2}, \quad \tilde{Z}_{1x,k}(\phi) = \frac{Z_{1x,k}(\phi)}{Z_{1,k}}, \quad \tilde{Z}_{1\tau,k}(\phi) = \frac{v_k^2}{Z_{1,k}} Z_{1\tau,k}(\phi),
$$
$$
\tilde{Z}_{2,k}(\phi) = \frac{Z_{2,k}(\phi)}{Z_{2,k}}, \quad \tilde{Z}_{3,k}(\phi) = \frac{v_k}{\sqrt{Z_{1,k} Z_{2,k}}} Z_{3,k}(\phi),
$$

(64)

where

$$
\langle \tilde{Z}_{1x,k}(\phi) \rangle_\phi = \langle \tilde{Z}_{2,k}(\phi) \rangle_\phi = 1, \qquad \langle \cdots \rangle_\phi = \frac{\sqrt{2}}{\pi} \int_{-\pi/2\sqrt{2}}^{\pi/2\sqrt{2}} d\phi \, (\cdots).
$$

(65)

The quantities $Z_{1\tau,k}$ and $Z_{3,k}$ introduced in Sec. 2.2 can be expressed as

$$
Z_{1\tau,k} = \langle \tilde{Z}_{1\tau,k}(\phi) \rangle, \qquad Z_{3,k} = \langle \tilde{Z}_{3,k}(\phi) \rangle.
$$

(66)

### A.2  Flow equations

The flow equations take the form

$$
\begin{aligned}
\partial_t \tilde{U}_k(\phi) &= (\eta_{1,k} - 2) \tilde{U}_k(\phi) + \mathcal{F}_U, \\
\partial_t \tilde{Z}_{1x,k}(\phi) &= \eta_{1,k} \tilde{Z}_{1x,k}(\phi) + \mathcal{F}_{Z_{1x}}, \\
\partial_t \tilde{Z}_{1\tau,k}(\phi) &= (2z_k - 2 + \eta_{1,k}) \tilde{Z}_{1\tau,k}(\phi) + \mathcal{F}_{Z_{1\tau}}, \\
\partial_t \tilde{Z}_{2,k}(\phi) &= \eta_{2,k} \tilde{Z}_{2,k}(\phi) + \mathcal{F}_{Z_2}, \\
\partial_t \tilde{Z}_{3,k}(\phi) &= \left( z_k - 1 + \frac{\eta_{1,k} + \eta_{2,k}}{2} \right) \tilde{Z}_{3,k}(\phi) + \mathcal{F}_{Z_3},
\end{aligned}
$$

(67)

where $\eta_{1,k} = -\partial_t \ln Z_{1,k}$, $\eta_{2,k} = -\partial_t \ln Z_{2,k}$ and $z_k = 1 + \partial_t \ln v_k$ is the dynamical exponent. The threshold functions $\mathcal{F}$ can be expressed as integrals over the dimensionless propagator $\tilde{G}_k = (\tilde{\Gamma}_k^{(2)} + \tilde{R}_k)^{-1}$ and depend on the dimensionless functions (64) and cutoff function $\tilde{R}_k$. The explicit form of the flow equations is too complicated to be shown here but it can be easily shown that they imply (18) and (19).

# B Gauge invariance

In the presence of an external gauge field $A = (A_0, A_x)$, the partition function reads

$$\mathcal{Z}[J, A] = \int \mathcal{D}[\varphi, \vartheta] \, e^{-S[\varphi, \vartheta, A] + \int_X J\varphi}, \tag{68}$$

where

$$S[\varphi, \vartheta, A] = \int_X \left\{ \frac{v}{2\pi} \left[ \frac{1}{K} (\partial_x \varphi^2) + K(\partial_x \vartheta - A_x)^2 \right] - \frac{i}{\pi} \partial_x \varphi (\partial_\tau \vartheta - A_0) - u \cos(2\sqrt{2}\varphi) \right\} \tag{69}$$

and $J$ is an external source that couples to $\varphi$. The current densities are defined by

$$
\begin{aligned}
J_0(X) &= -\frac{\delta S[\varphi, \vartheta, A]}{\delta A_0(X)} = -\frac{i}{\pi} \partial_x \varphi(X), \\
J_x(X) &= -\frac{\delta S[\varphi, \vartheta, A]}{\delta A_x(X)} = \frac{vK}{\pi} [\partial_x \vartheta(X) - A_x(X)].
\end{aligned} \tag{70}
$$

When the source $J(X) = J$ is static and uniform, the expectation value $\langle J_\mu(X) \rangle$ vanishes for $A_\mu = 0$. The linear response to the gauge field is given by

$$\langle J_\mu(X) \rangle = \int_{X'} K_{\mu\nu}(X, X', J) A_\nu(X') + \mathcal{O}(A^2) \tag{71}$$

(with an implicit sum over repeated discrete indices), where

$$K_{\mu\nu}(X, X', J) = \frac{\delta^2 \ln \mathcal{Z}[J, A]}{\delta A_\mu(X) \delta A_\nu(X')} \bigg|_{A=0} = \Pi_{\mu\nu}(X, X', J) - \frac{vK}{\pi} \delta_{\mu,x} \delta_{\nu,x}, \tag{72}$$

and

$$\Pi_{\mu\nu}(X, X', J) = \langle j_\mu(X) j_\nu(X') \rangle \tag{73}$$

is the correlation function of the paramagnetic part of the current density: $j_x = (vK/\pi)\partial_x \vartheta$ and $j_0 = J_0 = -(i/\pi)\partial_x \varphi$.

For a pure gauge field, $A_\mu = \partial_\mu \xi$ (with $\xi(X)$ an arbitrary function), the expectation value of the current densities must vanish, i.e.

$$0 = \int_{X'} K_{\mu\nu}(X, X', J) \partial_{X'_\nu} \xi(X') = -\int_{X'} [\partial_{X'_\nu} K_{\mu\nu}(X, X', J)] \xi(X'). \tag{74}$$

Equation (74) implies that the electromagnetic response function is transverse,

$$\partial_{X_\mu} K_{\mu\nu}(X, X', \phi) = \partial_{X'_\nu} K_{\mu\nu}(X, X', \phi) = 0, \tag{75}$$

where we now consider $K_{\mu\nu}$ as a function of the (constant) field $\phi = \langle \varphi(X) \rangle$ rather then the source $J$. In Fourier space, Eq. (75) gives

$$-\omega K_{0\nu}(Q, \phi) + q K_{x\nu}(Q, \phi) = 0. \tag{76}$$

Using Eqs. (72,73) and the expression of $j_\mu$, we finally obtain

$$
\begin{aligned}
\omega G_{\varphi\varphi}(Q, \phi) - ivKq G_{\varphi\vartheta}(Q, \phi) &= 0, \\
i\omega q G_{\varphi\vartheta}(Q, \phi) + vKq^2 G_{\vartheta\vartheta}(Q, \phi) - \pi &= 0,
\end{aligned} \tag{77}
$$

or, equivalently,

$$\omega \Gamma^{(2)}_{\theta\theta}(Q,\phi) + iv Kq\Gamma^{(2)}_{\phi\theta}(Q,\phi) = 0,$$
$$-i\omega q\Gamma^{(2)}_{\phi\theta}(Q,\phi) + v Kq^2\Gamma^{(2)}_{\phi\phi}(Q,\phi) - \pi\det\Gamma^{(2)}(Q,\phi) = 0. \tag{78}$$

Let us now consider the most general expression of the two-point vertex

$$\Gamma^{(2)}_k(Q,\phi) = \begin{pmatrix} Z_{1x,k}(\phi)q^2 + Z_{1\tau,k}(\phi)\omega^2 + U_k''(\phi) & iZ_{3,k}(\phi)q\omega \\ iZ_{3,k}(\phi)q\omega & Z_{2,k}(\phi)q^2 + Z_{2\tau,k}(\phi)\omega^2 \end{pmatrix} \tag{79}$$

to second order in $q$ and $\omega$ and compatible with symmetries.[5] Equation (79) would be obtained from the full effective action to second order in the derivative expansion, i.e. including a term $\frac{1}{2}Z_{2\tau,k}(\phi)(\partial_\tau\theta)^2$. From (78) and (79) we deduce

$$\lim_{Q\to 0}\frac{\partial}{\partial q^2}\Gamma^{(2)}_{\theta\theta}(Q,\phi) = Z_{2,k}(\phi) = \frac{vK}{\pi},$$
$$\lim_{Q\to 0}\frac{\partial}{\partial\omega^2}\Gamma^{(2)}_{\theta\theta}(Q,\phi) = Z_{2\tau,k}(\phi) = 0, \tag{80}$$
$$\lim_{Q\to 0}\frac{\partial^2}{\partial q\partial\omega}\Gamma^{(2)}_{\phi\theta}(Q,\phi) = Z_{3,k}(\phi) = \frac{i}{\pi}.$$

Gauge invariance implies that $Z_{2,k}(\phi)$, $Z_{2\tau,k}(\phi)$ and $Z_{3,k}(\phi)$ are not renormalized and remain equal to their initial value.

## C  Flow equations of the sine-Gordon model

All properties related to the density field $\varphi$ can be obtained from the effective potential $U_k(\phi)$ and the propagator

$$G_{\varphi\varphi,k}(Q,\phi) = \frac{1}{Z_{1x,k}(\phi)(q^2 + \omega^2/v^2) + U_k''(\phi)}. \tag{81}$$

The equation for the derivative of the effective potential reads

$$\partial_t U_k'(\phi) = -\frac{1}{2}\int_Q G_{1i,k}(Q,\phi)\partial_t R_{ij,k}(Q)G_{j1,k}(Q,\phi)\Gamma^{(3)}_{111,k}(0,Q,-Q) \tag{82}$$

($\partial_t = k\partial_k$ and an implicit sum over discrete indices $i,j = 1,2$ is assumed), where we assign the index 1 to $\phi$ and 2 to $\theta$ and use the notation $\int_Q = \int\frac{dq}{2\pi}\int\frac{d\omega}{2\pi}$. $G_{11,k}(Q,\phi)$ is given by (81) and

$$G_{12,k}(Q,\phi) = -\frac{i}{vK}\frac{\omega/q}{Z_{1x,k}(\phi)(q^2 + \omega^2/v^2) + U_k''(\phi)}. \tag{83}$$

Equation (82) is obtained by noting that the only nonzero three-point vertex has all external legs corresponding to $\phi$, its expression is identical to that in the sine-Gordon model. Using

$$\partial_t R_k(Q) = \begin{pmatrix} -\eta_{1,k}Z_{1,k}Q^2 r(\tilde{Q}^2) & 0 \\ 0 & 0 \end{pmatrix} - 2\tilde{Q}^2 r'(\tilde{Q}^2)\begin{pmatrix} Z_{1k}Q^2 - \frac{\omega^2}{\pi vK} & \frac{i}{\pi}q\omega \\ \frac{i}{\pi}q\omega & \frac{vK}{\pi}q^2 \end{pmatrix}, \tag{84}$$

where $\eta_{1,k} = -\partial_t\ln Z_{1,k}$, $Q^2 = q^2 + \omega^2/v^2$ and $\tilde{Q}^2 = Q^2/k^2$, we obtain

$$G_{1i,k}(Q,\phi)\partial_t R_{ij,k}(Q)G_{j1,k}(Q,\phi) = -Z_{1,k}Q^2[\eta_{1,k}r(\tilde{Q}^2) + 2\tilde{Q}^2 r'(\tilde{Q})]G_{11,k}(Q)^2 \tag{85}$$

and therefore

$$\partial_t U_k'(\phi) = \frac{1}{2}\int_Q Z_{1,k}Q^2[\eta_{1,k}r(\tilde{Q}^2)+2\tilde{Q}^2 r'(\tilde{Q}^2)]G_{11,k}(Q)^2\Gamma_{111,k}^{(3)}(0,Q,-Q). \tag{86}$$

This equation coincides with the flow equation of the effective potential in the sine-Gordon model.

By a similar reasoning we can show that the flow equation for $Z_{1x,k}(\phi)$, which is deduced from $\partial_t\Gamma_{11,k}^{(2)}(Q)$, is identical to the equation derived within the sine-Gordon model. This simply follows from the two following properties: i) the vertices $\Gamma_{111,k}^{(3)}$ and $\Gamma_{1111,k}^{(4)}$ (the only three- and four-point vertices that are nonzero) are the same as in the sine-Gordon model, ii) the rhs of (85) is equal to the quantity $\partial_t R_k(Q)G_k(Q)^2$ in the sine-Gordon model. As long as all external legs correspond to the field $\phi$, the propagator $G_{22,k}$ does not appear in the flow equation and $G_{12,k}$ enters only *via* Eq. (85). Using the latter amounts to integrating out the field $\vartheta$ at the level of the flow equations.

## D Flow equation $\partial_k\bar{\Gamma}_k[\phi,\bar{\theta}]$

It is convenient to use the notation $\varphi_1 = \varphi_{1,k} = \varphi$, $\varphi_2 = \vartheta$, $\varphi_{2,k} = \bar{\vartheta}_k$ and introduce the two-component fields

$$\bar{\varphi}_k \equiv \bar{\varphi}_k[\varphi,\vartheta] = \begin{pmatrix} \varphi \\ \bar{\vartheta}_k[\varphi,\vartheta] \end{pmatrix}, \qquad \bar{\Phi} = \langle\bar{\varphi}_k\rangle = \begin{pmatrix} \phi \\ \bar{\theta} \end{pmatrix}. \tag{87}$$

The scale-dependent effective action $\bar{\Gamma}_k[\bar{\Phi}]$ is defined by (60) and satisfies the equation of motion

$$\frac{\delta\bar{\Gamma}_k[\bar{\Phi}]}{\delta\bar{\Phi}_i(X)} = J_i(X) - \int_Y \bar{R}_{ij,k}(X,Y)\bar{\Phi}_j(Y) \tag{88}$$

as well as

$$\bar{\Gamma}_k^{(2)} + \bar{R}_k = \mathcal{W}_k^{(2)-1}, \tag{89}$$

where $J = (J_1,J_2)^T \equiv (J_\varphi, J_{\bar{\vartheta}})^T$ and $\mathcal{W}_k^{(2)}[J]$ is the second-order functional derivative of $\mathcal{W}_k[J] = \ln\mathcal{Z}_k[J]$.

To derive the flow equation we start from

$$\partial_k\mathcal{W}_k[J] = -\frac{1}{2}\int_{X,Y}\partial_k R_{ij,k}(X,Y)\langle\varphi_j(Y)\varphi_i(X)\rangle + \int_X J_i\langle\partial_k\bar{\varphi}_{i,k}\rangle \tag{90}$$

and

$$\partial_k\bar{\Gamma}_k[\bar{\Phi}] = \frac{1}{2}\int_{X,Y}\partial_k R_{ij,k}(X,Y)\langle\varphi_j(Y)\varphi_i(X)\rangle - \int_X J_i\langle\partial_k\bar{\varphi}_{i,k}\rangle - \partial_k\Delta\bar{S}_k[\bar{\Phi}]. \tag{91}$$

The $k$ derivative is taken at fixed source $J$ in (90) and at fixed field $\bar{\Phi}$ in (91) and we have used $\bar{\Phi}_i(X) = \delta\mathcal{W}_k[J]/\delta J_i(X)$ to obtain (91). Since

$$\partial_k\Delta\bar{S}_k[\bar{\Phi}]\Big|_{\bar{\Phi}} = \frac{1}{2}\int_{X,Y}\bar{\Phi}_i(X)\partial_k\bar{R}_{ij,k}(X,Y)\bar{\Phi}_j(Y),$$

$$\int_X J_i\langle\partial_k\bar{\varphi}_{i,k}\rangle = \int_X\left[\frac{\delta\bar{\Gamma}[\bar{\Phi}]}{\delta\bar{\Phi}_i(X)} + \int_Y\bar{R}_{ij,k}(X,Y)\bar{\Phi}_j(Y)\right]\langle\partial_k\bar{\varphi}_{i,k}(X)\rangle \tag{92}$$

and

$$\frac{1}{2}\int_{X,Y}\partial_k R_{ij,k}(X,Y)\langle\varphi_j(Y)\varphi_i(X)\rangle = \frac{1}{2}\partial_k\int_{X,Y}\langle\bar\varphi_{i,k}(X)\bar R_{ij,k}(X,Y)\bar\varphi_{j,k}(Y)\rangle$$

$$= \int_{X,Y}\left\{\frac{1}{2}\partial_k\bar R_{ij,k}(X,Y)\langle\bar\varphi_{i,k}(X)\bar\varphi_{j,k}(Y)\rangle\right.$$

$$\left.+\bar R_{ij,k}(X,Y)\langle\partial_k\bar\varphi_{i,k}(X)\bar\varphi_{j,k}(Y)\rangle\right\}, \tag{93}$$

we finally deduce

$$\partial_k\bar\Gamma_k[\bar\Phi] = \frac{1}{2}\mathrm{Tr}\left[\partial_k\bar R_k\big(\bar\Gamma_k^{(2)}[\bar\Phi]+\bar R_k\big)^{-1}\right] - \int_X\frac{\delta\bar\Gamma_k[\bar\Phi]}{\delta\bar\Phi_i(X)}\langle\partial_k\bar\varphi_{i,k}(X)\rangle$$

$$+\int_{X,Y}\bar R_{ij,k}(X,Y)[\langle\partial_k\bar\varphi_{i,k}(X)\bar\varphi_{j,k}(Y)\rangle-\langle\partial_k\bar\varphi_{i,k}(X)\rangle\bar\Phi_j(Y)]. \tag{94}$$

This equation can be rewritten as in (62).

# E  $\Gamma_k[\phi,\theta_k[\phi,\bar\theta]]$ vs $\bar\Gamma_k[\phi,\bar\theta]$

Since the effective actions $\Gamma_k[\phi,\theta_k[\phi,\bar\theta]]$ and $\bar\Gamma_k[\phi,\bar\theta]$ satisfy the same initial condition at $k=k_{\mathrm{in}}$, they are identical if they satisfy the same flow equation, i.e. if

$$\partial_k\Gamma_k[\Phi_k[\bar\Phi]]\Big|_{\bar\Phi} = \frac{1}{2}\mathrm{Tr}\left[\partial_k R_k\big(\Gamma_k^{(2)}[\Phi_k[\bar\Phi]]+R_k\big)^{-1}\right] + \int_X\frac{\delta\Gamma_k[\Phi]}{\delta\Phi_i(X)}\bigg|_{\Phi=\Phi_k[\bar\Phi]}\partial_k\Phi_i(X)\Big|_{\bar\Phi} \tag{95}$$

coincides with $\partial_k\bar\Gamma_k[\bar\Phi]$. Here $\Phi=(\phi,\theta)^T\equiv\Phi_k[\bar\Phi]]$ is considered as a $k$-dependent functional of $\bar\Phi$. It is convenient to write the relation between $\Phi$ and $\bar\Phi$ as

$$\Phi(Q) = M_k(Q)\bar\Phi(Q) \quad\text{with}\quad M_k(Q) = \begin{pmatrix} 1 & 0 \\ \alpha_k(Q) & \beta_k(Q) \end{pmatrix} \tag{96}$$

so that

$$\bar R_k = M_k^T R_k M_k, \qquad G_k = M_k\bar G_k M_k^T, \tag{97}$$

where $G_k=(\Gamma_k^{(2)}[\Phi]+R_k)^{-1}$ and $\bar G_k=(\bar\Gamma_k^{(2)}[\bar\Phi]+\bar R_k)^{-1}$. One then easily finds

$$\frac{1}{2}\mathrm{Tr}(\partial_k R_k G_k) = \frac{1}{2}\mathrm{Tr}(\partial_k\bar R_k\bar G_k - 2\bar R_k M_k^{-1}\partial_k M_k\bar G_k)$$

$$= \frac{1}{2}\mathrm{Tr}(\partial_k\bar R_k\bar G_k) + \int_{X,Y}\bar R_{ij,k}(X,Y)\langle\partial_k\bar\varphi_{i,k}(X)\bar\varphi_{j,k}(Y)\rangle_c. \tag{98}$$

Assuming that $\Gamma_k[\Phi_k[\bar\Phi]]=\bar\Gamma_k[\bar\Phi]$ holds at scale $k$, one obtains

$$\int_X\frac{\delta\Gamma_k[\Phi]}{\delta\Phi_i(X)}\bigg|_{\Phi=\Phi_k[\bar\Phi]}\partial_k\Phi_i(X)\Big|_{\bar\Phi} = \sum_Q\frac{\delta\Gamma_k[\Phi]}{\delta\Phi_i(Q)}\bigg|_{\Phi=\Phi_k[\bar\Phi]}\partial_k\Phi_i(Q)\Big|_{\bar\Phi}$$

$$= \sum_Q\frac{\delta\bar\Gamma_k[\Phi]}{\delta\bar\Phi_j(Q)}M_{ji,k}^{-1}(Q)\partial_k M_{il,k}(Q)\bar\Phi_l(Q)$$

$$= -\int_X\frac{\delta\bar\Gamma_k[\bar\Phi]}{\delta\bar\Phi_i(X)}\langle\partial_k\bar\varphi_{i,k}(X)\rangle. \tag{99}$$

From Eqs. (94) and (95,98,99) we deduce

$$\partial_k\Gamma_k[\Phi_k[\bar\Phi]]\Big|_{\bar\Phi} = \partial_k\bar\Gamma_k[\bar\Phi]\Big|_{\bar\Phi}. \tag{100}$$

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
