# Peer review of "Flowing bosonization in the nonperturbative functional renormalization-group approach"

_SciPost Physics, doi:SciPost Phys. 12, 110 (2022)_

## Round 2 · Referee Report · Anonymous · 2022-1-14

Strengths
1-original work
2-application of a method (dynamical bosonization) developed for other fields of physics to 1D quantum fluids.
3-manuscript well written and clear
Weaknesses
1-potential applications of the method are not sufficiently presented in the manuscript
Report
In this paper, the authors develop an extension of the nonperturbative functional renormalization group (FRG) for one-dimensional (1D) quantum fluids. In particular, they analyze a bosonized theory described by two fields: the "density" field $\varphi$ and its conjugate, the superfluid phase $\vartheta$. The main result is that the phase field has to be dynamically redefined for $\varphi$ and $\vartheta$ to be conjugate variables at any FRG scale $k$. This "flowing bosonization" procedure is then applied to the case of a Luttinger liquid in presence of a periodic potential.
I believe this paper to be well written, clear, and methodologically interesting.
I have nevertheless some minor comments I would ask the authors to address.
1- Page 6: the authors write "We construct [...] $R_k( Q)$ in the usual way." I believe that here a citation to other works where this type of regulator has been used is needed.
2- Page 6: "[...] we take $r(y)=\alpha/(e^y-1)$ with $\alpha$ of order unity." I think a brief discussion on how $\alpha$ is chosen is necessary.
3- Page 10, below Eq.(37): I believe, for clarity sake, that the authors should stress at this stage that $\bar{K}_k$ is obtained from the average of $Z_{1,k}(\phi)$, while $K_k$ from its value at zero field.
4- Sec. 3.1.1: In my opinion it is more sound to write Eq.(38) as
$\theta(Q)=-i\omega \alpha_k(Q) \phi(Q) + \beta_k(Q) \bar{\theta}(Q)$
as it is clear that $\theta$ must be a combination of $\bar{\theta}$ and $\partial_\tau \phi$. In this way it is more evident that the term in the second line of (39) is a $(\partial_\tau\phi)^2$ term and it has to vanish when summed with $Z_{1\tau,k}(\phi=0)(\partial_\tau\phi)^2$.
5- Page 13: The meaning of the paragraph from "It was pointed out in Ref. [3] ... " to "... preventing the superfluid stiffness to vanish when $k\to 0$." is unclear to me. I would ask the authors to clarify.
6- Sec. 3.2, below Eq.(57): the authors write "The coefficients $\alpha$ and $\beta$ are given by (41)." It is not clear whether, within an active frame perspective, one has to assume the coefficients to be given by Eq.(41) or this can be somewhat derived in a similar way to what is done for the passive frame approach.
7- In the conclusion the authors refer to the possibility of studying the Bose fluid by directly using the bosonic fields $\psi$ and $\psi^*$. I believe this discussion deserves to be at least briefly mentioned in the introduction as well, as it seems to be a little decontextualized.
8- I believe that the authors should improve their conclusion by further stressing (if possible) the potential applications of their method.
9- Appendix A.2: I think, for completeness sake, that if the authors do not want to show the explicit form of their flow equations, they should at least refer to other works where they appear.
In summary, I believe this paper can be published on SciPost Physics, after the authors have considered the changes listed above.

---

## Round 2 · Referee Report · Anonymous · 2022-1-19

Strengths
1-Clear and detailed presentation
2-Methodological advance
Weaknesses
1-Only minor new physical insight
Report
This work discusses an implementation of the nonperturbative (or functional) renormalization group to one-dimensional quantum fluids within the partial bosonization approach. While previous works typically completely integrate out the phase field, resulting in an effective action for the density field alone, the approach discussed here keeps the phase field in the action. This allows one to study fluctuations of the phase field and might be applicable to more generic situations in which the action is no longer quadratic in the phase field. The approach is shown to reproduce the expected results for the Luttinger and Mott insulating phases, as long as a flowing (or dynamical) bosonization scheme is employed. Such schemes are by now widely used in the renormalization group literature in the context of high-energy, nuclear, and atomic physics problems, but have apparently so far not been employed in the description of one-dimensional quantum fluids. The paper is very well written and appears without obvious flaws. I believe that the result obtained here represent a significant advance as required for publication in SciPost Physics.
I do have two minimal comments that the authors might want to address before publication, see below.
Requested changes
1-As mentioned in the introduction and shown in Appendix B, the fact that the superfluid stiffness does not renormalize despite the presence of a periodic potential in the partially bosonized scheme without flowing bosonization, is a consequence of gauge invariance. It is also claimed that this statement holds independent of the approximation scheme used. I did not understand the latter statement. It seems to imply that it holds to arbitrary orders in the derivative expansion, while the actual calculations performed in Appendix B appear to make explicit use of the second-order truncation of the derivative expansion. Can the authors please elucidate this point?
2-It might be useful to repeat the definition of the RG time t in the captions of Fig. 1, as these are used in the labels of the plots.

---

## Round 3 · Author Response

Dear Editor,

we would like to thank the referees for their report on the manuscript, their positive judgment and their constructive remarks. We reply below to their comments and questions. A redlined version of the manuscript that highlights the changes and a pdf version of the letter are provided in the comments section.

Sincerely Yours,

Nicolas Dupuis and Romain Daviet

Report 1:

1-As mentioned in the introduction and shown in Appendix B, the fact that the superfluid stiffness does not renormalize despite the presence of a periodic potential in the partially bosonized scheme without flowing bosonization, is a consequence of gauge invariance. It is also claimed that this statement holds independent of the approximation scheme used. I did not understand the latter statement. It seems to imply that it holds to arbitrary orders in the derivative expansion, while the actual calculations performed in Appendix B appear to make explicit use of the second-order truncation of the derivative expansion. Can the authors please elucidate this point?

Answer: We thank the referee for drawing our attention to this issue. In fact Appendix B does not rely on a truncation of the effective action to second order in the derivative expansion but only makes use of the most general derivative expansion of the two-point vertex $\Gamma^{(2)}$ as given in Eq.~(79). We have changed the text above and below~(79) and modified Eqs.~(80) to clarify this point.

2-It might be useful to repeat the definition of the RG time t in the captions of Fig. 1, as these are used in the labels of the plots.

Answer: We thank the referee for this suggestion.

Report 2:

1- Page 6: the authors write "We construct [...] $R_k(Q)$ in the usual way." I believe that here a citation to other works where this type of regulator has been used is needed.

Answer: We have slightly modified the sentence and added two references.

2- Page 6: "[...] we take $ r(y)=\alpha/(e^y-1)$ with $\alpha$ of order unity." I think a brief discussion on how $\alpha$ is chosen is necessary.

Answer: In the case of a precision calculation (e.g. when determining the critical exponents at a second-order phase transition), $\alpha$ is fixed by using the principle of minimum sensitivity. In the sine-Gordon model, the precise value of $\alpha$ is unimportant (see Ref.~[2]). In the present study, it is therefore sufficient to take $\alpha$ of order unity. We have added a comment in the text.

3- Page 10, below Eq.(37): I believe, for clarity sake, that the authors should stress at this stage that $\tilde K_k$ is obtained from the average of $Z_{1k}(\phi)$, while $K_k$ from its value at zero field.

Answer: We thank the referee for this suggestion.

4- Sec. 3.1.1: In my opinion it is more sound to write Eq.(38) as $\theta(Q)=-i\omega \alpha_k(Q) \phi(Q) + \beta_k(Q) \bar\theta(Q)$ as it is clear that $\theta$ must be a combination of $\bar\theta$ and $\partial_\tau\phi$. In this way it is more evident that the term in the second line of (39) is a $(\partial_{\tau} \phi)^2$ term and it has to vanish when summed with $Z_{1\tau,k}(\phi=0)( \partial_{\tau} \phi)^2$.

Answer: Actually it is $\partial_x\theta$ which is a combination of $\partial_x\bar\theta$ and $\partial_\tau\phi$, so that in the end $\alpha_k(Q)\sim i\omega/q$ as obtained in Eq.~(41). But we agree with the referee that we could write~(38) as $\theta(Q)=-i(\omega/q) \alpha_k(Q) \phi(Q) + \beta_k(Q) \bar\theta(Q)$. On the other hand our notations are well suited to the calculations of Appendix~E since they give a simple expression of the matrix $M_k(Q)$. For this reason we prefer keep Eq.~(38) as it is now.

5- Page 13: The meaning of the paragraph from "It was pointed out in Ref. [3] ... " to "... preventing the superfluid stiffness to vanish when $k\to 0$." is unclear to me. I would ask the authors to clarify.

Answer: What we want to explain here is the fact that the convergence of $\eta_{1,k}=-\dt\ln Z_{1,k}$ towards 2 makes the regulator of order $Z_{1,k}k^2\sim k^{2-\eta_k}$ for $|q|,|\omega|$ of order $k$. Thus the convergence of $\eta_{1,k}$ towards 2 must be extremely slow, which is not realized in practice, for the regulator function $R_k$ to vanish in the infrared. While this issue is irrelevant for most physical quantities, which rapidly converge when $k$ becomes smaller than the mass scale $m_k/v$, the non-vanishing of $R_k$ may artificially stop the flow of $K_k$ thus preventing the superfluid stiffness to vanish when $k\to 0$. We have improved the discussion of this issue in the manuscript.

6- Sec. 3.2, below Eq.(57): the authors write "The coefficients $\alpha$ and $\beta$ are given by (41)." It is not clear whether, within an active frame perspective, one has to assume the coefficients to be given by Eq.(41) or this can be somewhat derived in a similar way to what is done for the passive frame approach.

Answer: This is a consequence of the change of variables being linear as shown in Ref.~[21]. We have slightly changed the text after Eq.~(57) and added a reference to~[21].

7- In the conclusion the authors refer to the possibility of studying the Bose fluid by directly using the bosonic fields $\psi$ and $\psi^*$. I believe this discussion deserves to be at least briefly mentioned in the introduction as well, as it seems to be a little decontextualized.

Answer: We have followed the referee's suggestion and added a paragraph in the introduction.

8- I believe that the authors should improve their conclusion by further stressing (if possible) the potential applications of their method.

Answer: We have slightly expanded the discussion on the two potential applications of our method in the conclusion: the Bose-glass phase of one-dimensional disordered bosons and the systems of weakly coupled one-dimensional chains.

9- Appendix A.2: I think, for completeness sake, that if the authors do not want to show the explicit form of their flow equations, they should at least refer to other works where they appear.

Answer: The flow equations did not appear elsewhere as they are specific to the two-field formalism used in the manuscript. We could show explicitly the flow equations in the Appendix (it would take at least two pages) or provide a mathematica file but we are not sure that this would be very illuminating. If the referee thinks otherwise, of course we will be happy to provide the file as a supplemental material. The flow equations are shown in a separate pdf file (see comments section below).

---

## Round 3 · List of Changes

Introduction:

Added "The preservation of the canonical commutation relations between $\partial_{x} \varphi$ and $\vartheta$ along the RG flow turns out to be crucial for a proper physical description of the system, in particular for the identification of the stiffness of the phase $\vartheta$ as the superfluid density. This differs from the study of a Bose fluid in a periodic potential using the canonically conjugated variables defined by the creation of annihilation boson fields. In that case, the fields $\psi$ and $\psi^*$ defined at the microscopic scale yield a simple identification of the superfluid density in the low-energy limit even though their canonical commutation relations are not preserved along the RG flow (see, e.g., [11,12] for an FRG study of the Bose-Hubbard model in two and three dimensions)."

Page 6 added :" We construct the regulator function $R_k(Q)$ by adapting the usual procedure [2,18] to the two-field formalism used here"

Page 6 added: "In the case of a precision calculation, e.g. when determining critical exponents at a second-order phase transition [33,34], $\alpha$ is fixed by using the principle of minimal sensitivity. In the sine-Gordon model, the precise value of $\alpha$ is unimportant [2]. In the present study it is therefore sufficient to take $\alpha$ of order unity."

Caption figure 1, added:"In Figs. 1 and 2, $t=\ln(k/\Lambda)$ denotes the (negative) RG time."

Page 10 added: "the field average of $Z_{1x,k}(\phi)$, i.e. $Z_{1,k}=v/\pi\bar K_k$"

Page 12 modified discussion: "A possible explanation comes from the convergence of $\eta_{1,k}=-\dt\ln Z_{1,k}$ towards 2 which makes the regulator of order $Z_{1,k}k^2\sim k^{2-\eta_k}$ for $|q|,|\omega|$ of order $k$. Thus the convergence of $\eta_{1,k}$ towards 2 must be extremely slow, which is not realized in practice, for the regulator function $R_k$ to vanish in the infrared [3]."

Page 14:" For a linear change of variables, the active frame transformation~(56) is the counterpart of the passive transformation~(38) and the coefficients $\alpha_k(Q)$ and $\beta_k(Q)$ are therefore given by (41) [25]."

Conclusion, modification:"For instance, this will allow a more accurate study of the Bose-glass phase of a one-dimensional disordered Bose fluid. The previous works using bosonization and FRG are based on an effective model obtained by integrating out the field $\vartheta$ from the outset [4,5]. This is sufficient to determine the properties related to the density field and its fluctuations but provides us with little information on the superfluid properties and the correlation function of the phase field $\vartheta$.
The work reported in this manuscript also opens up the possibility to study strongly anisotropic two- or three-dimensional systems, consisting of weakly coupled one-dimensional chains. In these systems the interchain kinetic coupling $\psi^*_{n}\psi_m \sim e^{-i\vartheta_n+i\vartheta_m}$ depends nontrivially on $\vartheta$ and it is not possible to integrate out this field from the outset. An RG approach must therefore necessarily consider the fields $\varphi$ and $\vartheta$ on equal footing."

Appendix B : modified discussion.

You are currently on this page

Resubmission 2111.11458v3 on 11 February 2022

---

## Editorial Decision

published